# Enhanced PeriOperative Care and Health protection programme for the prevention of surgical site infections after elective abdominal surgery (EPOCH): study protocol of a randomised controlled, multicentre, superiority trial

Stijn W de Jonge,[1] Niels Wolfhagen ![ORCID] ,[1] Quirine JJ Boldingh,[1] Wouter J Bom,[1] Linda M Posthuma,[2] Jochem CG Scheijmans,[1] Bart MF van der Leeuw,[3] Joost AB van der Hoeven,[4] Jens Peter Hering,[5] Dirk JA Sonneveld,[6] Otto E van Geffen,[7] Eduard R Hendriks,[8] Ewoud B Kluyver,[9] Ahmet Demirkiran,[10] Luc RCW van Lonkhuijzen,[11] Thomas Slotema,[12] Werner A Draaisma,[13] Seppe JSHA Koopman,[14] Charles C van Rossem,[15] Linda M Over,[16] Peter van Duijvendijk,[17] Marcel GW Dijkgraaf,[18] Markus W Hollmann,[2] Marja A Boermeester[1]

For numbered affiliations see end of article.

**Correspondence to**
Dr Marja A Boermeester;
m.a.boermeester@
amsterdamumc.nl

## ABSTRACT

**Introduction** Surgical site infections (SSI) are a common postoperative complication. During the development of the new WHO guidelines on SSI prevention, also in the Netherlands was concluded that perioperative care could be optimised beyond the current standard practice. We selected a limited set of readily available, cheap and evidence-based interventions from these new guidelines that are not part of standard practice in the Netherlands and formulated an Enhanced PeriOperative Care and Health bundle (EPOCH). Here, we describe the protocol for an open-label, randomised controlled, parallel-group, superiority trial to test the effect of the EPOCH bundle added to (national) standard care in comparison to standard care alone on the incidence of SSI.

**Methods and analysis** EPOCH consists of intraoperative high fractional inspired oxygen (0.80); goal-directed fluid therapy; active preoperative, intraoperative and postoperative warming; perioperative glucose control and treatment of severe hyperglycaemia (>10 mmol$^{l-1}$) and standardised surgical site handling. Patients scheduled for elective abdominal surgery with an incision larger than 5 cm are eligible for inclusion. Participants are randomised daily, 1:1 according to variable block sizes, and stratified per participating centre to either EPOCH added to standard care or standard care only. The primary endpoint will be SSI incidence according to the Centers for Disease Control and Prevention (CDC) definition within 30 days as part of routine clinical follow-up. Four additional questionnaires will be sent out over the course of 90 days to capture disability and costs. Other secondary endpoints include anastomotic leakage, incidence of incisional hernia, serious adverse events, hospital readmissions, length of stay and cost effectiveness. Analysis of the primary endpoint will be on an intention-to-treat basis.

**Ethics and dissemination** Ethics approval is granted by the Amsterdam UMC Medical Ethics Committee (reference 2015_121). Results will be disseminated through peer-reviewed journals and summaries shared with stakeholders. This protocol is published before analysis of the results.

**Trial registration number** Registered in the Dutch Trial Register: NL5572.

## Strengths and limitations of this study

► The intervention under investigation is a bundle of readily available, cheap and evidence-based interventions to reduce surgical site infection rates.
► A randomised controlled trial is used to test the effectiveness of the intervention bundle.
► Because the study investigates a bundle of interventions, inference on the attributive effect of each separate intervention of the bundle is challenging.
► Subjects and researchers cannot be blinded, but outcome assessors are unaware of treatment allocation.

## INTRODUCTION
### Background and rationale
Surgical site infections (SSI) cause excess morbidity, mortality, prolonged hospital stay

**Table 1** Overview of EPOCH interventions and typical standard care

| EPOCH interventions* | (Expected) Typical standard care |
|---|---|
| Intraoperative administration of high $FiO_2$: Patients will be administered a $FiO_2$ of 80% after intubation during the entire procedure until extubation. | Intraoperative administration of a $FiO_2$ of 30%–40%. |
| Goal-directed fluid therapy: Patients will receive haemodynamic therapy based on a goal-directed approach. Optimisation is preferably based on dynamic preload parameters (pulse pressure variation or systolic pressure variation) derived from arterial catheter measurements, when such catheter is used based on clinical indication. Arterial catheters will not be placed solely for the trial. When no arterial catheter is indicated, a goal-directed approach using a non-invasive haemodynamic monitoring system may be used. When both an arterial catheter and a non-invasive haemodynamic monitoring system are not available, hypotensive periods following induction should be compensated for by employing a vasopressive agent (norepinephrine) until a maximum of 0.08 gamma before fluids are administered. | Haemodynamic management based on clinical judgement including among other things heart rate, blood pressure and fluid balance. |
| Active preoperative, intraoperative and postoperative warming: Patients will be put under a hot air blanket (for example a bair hugger) from arrival at the preoperative holding, continued after arrival in the OR, until 2 hours after surgery in the recovery room. Core temperature is aimed at >36.5°C. If necessary additional legwarmers may be applied. Pre-warmed blankets, replaced when cooled down, can be used as an alternative when hot air blankets are not available. | Active intraoperative warming—but no preoperative and postoperative warming—. |
| Perioperative blood glucose control: All patients (diabetic and non-diabetic) will be subjected to blood glucose control throughout the procedure and on the first and second postoperative day. Blood glucose will be measured every hour during surgery (at least twice during procedures lasting more than 2 hours), once after surgery in the recovery room, and twice randomly (not fasting) on postoperative days 1 and 2. Blood glucose measurements over 10 $mmol^{l-1}$ will be treated with insulin. Success of glucose interventions will be checked. | Glucose control at discretion of the anaesthesiologist. |
| Surgical site handling: An alcohol-based antiseptic agent (chlorhexidine gluconate) will be used for skin preparation before incision. Before wound closure (after closure of the fascia), tissue will be irrigated with an aqueous antiseptic agent (0.35%–10% aqueous povidone iodine solution or 1% aqueous chlorhexidine gluconate) followed by lavage with normal saline. | No specific requirements for pre-incisional skin preparation agent or wound irrigation before closure |

*EPOCH interventions are supported by evidence-based WHO and CDC guideline recommendations.
CDC, Centers for Disease Control and Prevention; EPOCH, Enhanced PeriOperative Care and Health; $FiO_2$, fractional inspired oxygen; OR, operating room.

and consequently increased costs.[1] [2] In high-income countries, 9.4% of all patients undergoing gastrointestinal surgery suffer from SSI.[3] All SSI combined add up to the most frequently reported healthcare associated infection on admission.[4]

Approximately 55% of SSI are deemed preventable with evidence-based strategies.[5] However, individual measures alone have not resulted in a clinically relevant SSI reduction.[6–9] In contrast, efforts that have used systematic approaches, or bundles, have been successful to varying degrees.[10–12] The Institute for Healthcare Improvement developed the care-bundle concept as small set of evidence-based interventions to improve predefined clinical outcomes. When interventions are implemented as a bundle, better outcomes are reported compared with implementation of these same interventions individually.[13] Care bundles have been successfully used to reduce central venous catheter-line infection rates, ventilator-associated pneumonia rates and to improve the outcomes of patients presenting with sepsis.[13–16]

A national care bundle for SSI prevention (Postoperative Wound Infection or POWI bundle) was implemented in 2009 in the Netherlands.[17] The bundle comprises of

hygiene discipline (measured by operating theatre door movements), timing of antibiotic prophylaxis (between 15 and 60 min prior to incision), normothermia (aimed at a core temperature of 36.5°C postoperatively) and the avoidance of preoperative hair removal.[18] However, a clear national reduction in SSI has not been demonstrated since start.[19] Of these interventions, only normothermia is evidence-based.[20–23] For the other three interventions, there is either no evidence at all, or the available evidence indicates there is no strong association with SSI.[24–26]

The development of two new guidelines on SSI prevention by the WHO and the Centers for Disease Control and Prevention (CDC) highlighted that perioperative care can be optimised beyond this limited, four-interventions, Dutch POWI bundle.[27–30] Nonetheless these interventions have not been widely implemented yet. Considering current standards of practice, costs and the potential for implementation, we selected from these guidelines a limited set of readily available, cheap and evidence-based interventions that are not part of current national standard perioperative care and formulated an Enhanced PeriOperative Care and Health programme (EPOCH). EPOCH comprises of intraoperative high fractional

**Table 2** Outcomes of EPOCH trial

**Primary outcome**

| Outcome | Definition | Method of data collection |
|---|---|---|
| The incidence of surgical site infection within 30 days of follow-up after surgery. | SSI defined by the Centers for Disease Control and prevention (CDC)[31] | LHCR registry and medical chart review |
| **Secondary outcomes** | | |
| The incidence of surgical site infection within 90 days of follow-up after surgery. | SSI defined by the CDC[31] | LHCR registry and medical chart review |
| The incidence of surgical site infection within 30 days follow-up after surgery. | SSI defined through a post discharge self-assessment questionnaire[40 41] | Survey including a Dutch translation of the Bluebelle Wound Healing Questionnaire[40 41] |
| The incidence of surgical site infection within 30 days follow-up after surgery. | SSI defined by visual characteristics of definition of SSI of the CDC[31] | Survey including self-reported wound photos |
| The incidence of anastomotic leakage within 30 days follow-up after surgery. | Anastomotic leakage defined by the international study group for rectal cancer[42] | LHCR registry and medical chart review |
| The incidence of incisional hernia within 1 year follow-up after surgery. | Incisional hernia defined by the European Hernia Society, as any abdominal wall gap with or without bulge in the area of a postoperative scar perceptible or palpable by clinical examination or imaging[43 44] | LHCR registry and medical chart review |
| All-cause mortality within 1 year follow-up after surgery. | | Data retrieved from Dutch national registry (Basisregistratie Personen (BRP)) |
| Serious adverse events (SAE) within 30 days follow-up after surgery | SAE defined as any event that isfatal, threatens the life of the subject, makes hospital admission or an extension of the admission necessary, causespersistent or significant invalidity or work disability, manifests itself in acongenital abnormality or malformation and/or could, according to the trialmanagement team, have developed to a serious undesired medical event, butwas, however, prevented due to premature interference.[35] | LHCR registry and medical chart review |
| Length of hospital stay | Number of days between surgery and discharge, and the number of days of potential readmissions. | Data retrieved from medical chart |
| Related costs | Direct, indirect, medical and non-medical costs. Costs per quality-adjusted life year (QALY) constitute the primary health economic outcome. | Hospital healthcare utilisation retrieved from health informatics system. Survey on out-of-hospital healthcare utilisation and productivity losses from surveys containing the Institute for Medical Technology Assessment's (iMTA) Medical Consumption Questionnaire adjusted to the study setting and the iMTA Productivity Cost Questionnaire.[45]A survey with out-of-pocket expenses will be added to the survey |
| Hospital readmissions | Yes or no | Data extraction from medical chart |
| Health and disability | | Gathered with the WHO Disability Assessment Schedule 2.0 at 30, 60 and 90 days[46] |
| Health utility and QALY | | Gathered with the EQ-5D-3L at 30, 60 and 90 days[47] |
| **Additional secondary outcome, depending on additional funding** | | |
| All-cause mortality within 5 years follow-up after surgery. | | Data retrieved from Dutch national registry (BRP) |

EPOCH, Enhanced PeriOperative Care and Health; EQ-5D-5L, EuroQol 5-Dimensions 5-Levels; LHCR, Dutch National Surgical Complication Registry; SSI, surgical site infections.

| | STUDY PERIOD | | | | | | | | | |
| | Enrolment | Allocation | Post-allocation | | | | | | | |
| TIMEPOINT | Up to -1 day | -1 day | 0 | 1 day | 2 days | 10 days | 30 days | 60 days | 90 days | 1 year |
| **ENROLMENT:** | | | | | | | | | | |
| Eligibility screen | X | | | X | | | | | | |
| Informed consent | X | | | | | | | | | |
| Allocation | | X | | | | | | | | |
| **INTERVENTIONS:** | | | | | | | | | | |
| Surgical procedure standard care | | | X | | | | | | | |
| Surgical procedure standard care supplemented with EPOCH | | | X | X | X | | | | | |
| **ASSESSMENTS:** | | | | | | | | | | |
| Baseline characteristics | | | X | | | | | | | |
| Perioperative characteristics | | | X | | | | | | | |
| Adherence survey surgeon | | | X | | | | | | | |
| Adherence survey anesthesiologist | | | X | | | | | | | |
| Serious Adverse Event | | | As needed throughout protocol | | | | | | | |
| SSI by medical chart review | | | | | | | X | | X | |
| Incidence SSI by survey | | | | | | | X | | | |
| Anastomotic leakage | | | | | | | X | | X | |
| Incisional hernia | | | | | | | X | | | X |
| EQ-5D-3L | | | | | | X | X | X | X | |
| Wound photo | | | | | | X | X | | | |
| WHODAS 2.0 | | | | | | | X | X | X | |
| IMTA-MCQ | | | | | | | | | X | |
| IMTA-PCQ | | | | | | | | | X | |

**Figure 1**  Schedule of enrolment, interventions and assessments. EPOCH, EnhancedPeriOperative Care and Health; EQ-5D-5L, EuroQol 5-Dimensions 5-Levels; iMTA, Institute forMedical Technology Assessment; MCQ, MedicalConsumption Questionnaire; PCQ, ProductivityCost Questionnaire; SSI, surgical siteinfections; WHODAS, WHO Disability Assessment Schedule.

de Jonge SW, *et al. BMJ Open* 2020;**10**:e038196. doi:10.1136/bmjopen-2020-038196

**Table 3** Revision chronology EPOCH study protocol

**Revision chronology EPOCH study protocol**

| Version no. | Date | Main reasons for change |
|---|---|---|
| 1. | 28-03-2015 | NA, first submission to MEC |
| 2. | 14-07-2015 | Modifications requested by MEC d.d. 09-05-2015 |
| 3. | 28-07-2015 | The use of sutures, instead of staples, for skin closure is no longer required in the intervention group. Evidence emerged before the study start that rebuked earlier perceptions on the risk of SSI after the use of staples. |
| 4. | 12-11-2015 | Administrative modification requested by MEC d.d. 12-11-2015 to comply with renewed law for human research. |
| 5. | 22-02-2016 | Addition of patient generated wound photos to data collection. |
| 6. | 05-07-2018 | Addition of a post discharge SSI surveillance checklist at 30 days postoperatively to outcome data for all trial participants and tissue oxygen tension measurement and immunologic analysis to investigate mechanism of potential effect of the interventions for a limited subset of trial participants. Modification of SAE reporting from individual event reports to a list of all events every 6 months to alleviate administrative pressure. |
| 7. | 24-01-2019 | Modification of technique for tissue oxygenation measurement and immunologic analysis before first measurement was conducted. |
| 8. | 26-05-2019 | Modification of DSMB charter at request of the DSMB and modifications of the wording of inclusion criteria and replacement of non-evaluable patients to better reflect practice, and practical modifications to ease administrative burden and logistics. |
| 9. | 23-08-2019 | Modification of sample size and inclusion criteria for additional measurements in limited subset of trial participants. |
| 10 | 17-12-2019 | Modification of exclusion criteria |

DSMB, Data and Safety Monitoring Board; EPOCH, Enhanced PeriOperative Care and Health; MEC, Medical Ethics Committee; NA, not available; SAE, serious adverse events; SSI, surgical site infections.

inspired oxygen ($FiO_2$); intraoperative goal-directed fluid therapy (GDFT); active preoperative, intraoperative and postoperative warming; glucose control and treatment of hyperglycaemia (>10 mmol$^{l-1}$); in diabetic as well as non-diabetic patients and wound irrigation before closure using an aqueous antiseptic. Intraoperative high $FiO_2$, GDFT and active perioperative warming are expected to increase tissue perfusion and hence stimulate wound healing.

From a pragmatic perspective, as control to the EPOCH bundle elements, we chose standard care by discretion of the surgeon and anaesthesiologist as the comparator to the same standard care, supplemented with EPOCH. EPOCH interventions and contrasting typical standard care are listed in table 1.

### Objectives

Our primary objective is to determine if the EPOCH programme added to standard care is superior to standard care with respect to the incidence of SSI as defined by the CDC within 30 days post-surgery in patients undergoing elective abdominal surgery with abdominal incisions larger than 5 cm. We hypothesise that the EPOCH programme is superior to standard care alone, for the prevention of SSI.[31] Secondary objectives are described in table 2.

### Trial design

The EPOCH trial is designed as an open-label, pragmatic 1:1, randomised controlled, parallel-group, multicentre, superiority trial.

## METHODS AND ANALYSIS
### Study setting

The trial will be conducted in a multicentre setting involving academic, top-clinical and general hospitals throughout the Netherlands. A list of all participating centres and anticipated inclusions will be kept up to date with the Dutch central commission on human research (CCMO) on www.toetsingonline.nl (dossier number: NL52796.018.15). All study sites are expected to be able to perform the EPOCH interventions. When needed, the anaesthesiology or surgery expert on the Trial Steering Committee will instruct local clinicians before study initiation.

### Eligibility criteria
#### Inclusion criteria

In order to be eligible to participate in this study, a subject must meet the following criteria:

► Adult patients.
► Scheduled for elective abdominal surgery involving an abdominal incision larger than 5 cm (this includes

**Table 4** WHO trial registration data set

| WHO trial registration data set | |
|---|---|
| Primary registry and trial identifying number | Trial NL5572 (NTR5694) |
| Date of registration in primary registry | 2016-03-03 |
| Secondary identifying numbers | NA |
| Source(s) of monetary or material support | ZonMW Dutch Healthcare efficacy programme<br>Innovation fund Dutch healthcare insurers<br>Ethicon, Johnson & Johnson |
| Primary sponsor | Amsterdam University Medical Centre, Location AMC |
| Secondary sponsor(s) | NA |
| Contact for public queries | m.a.boermeester@amsterdamumc.nl |
| Contact for scientific queries | m.a.boermeester@amsterdamumc.nl |
| Public title | Enhanced care around surgery for the prevention of surgical site infections |
| Scientific title | Enhanced perioperative care for the prevention of surgical site infections, A pragmatic, randomised controlled, parallel-group, multicentre, superiority trial |
| Countries of recruitment | The Netherlands |
| Health condition(s) or problem(s) studied | Surgical site infection |
| Intervention(s) | Intervention:<br>An enhanced perioperative care programme added to standard care consisting of:<br>▶ Supplemental oxygen ($FiO_2$ 0.80)<br>▶ Preoperative, intraoperative and postoperative warming<br>▶ Goal-directed fluid therapy<br>▶ Perioperative glucose control <10 mmol$^{l-1}$<br>▶ Surgical wound irrigation<br>Control:<br>Standard care at discretion of treating physician |
| Key inclusion and exclusion criteria | Inclusion criteria:<br>Adult patients undergoing elective open or laparoscopic abdominal surgery with an incision larger than 5 cm.<br>Exclusion criteria:<br>▶ The need for emergency surgery<br>▶ Scheduled operation concerning a reoperation for complications from recent surgery (within 3 months after the initial procedure)<br>▶ Scheduled operation as part of a two staged surgical procedure and second procedure possibly scheduled before follow-up of primary endpoint<br>▶ The inability to read or understand informed consent material or study questionnaires<br>▶ Participation in another study with interference of study interventions or outcomes<br>▶ Pregnancy |
| Study type | A pragmatic, randomised controlled, parallel-group, multicentre, superiority trial |
| Date of first enrolment | 2016-03-01 |
| Target sample size | 3000 |
| Recruitment status | Recruiting |
| Primary outcome(s) | Incidence of surgical site infections within 30 days evaluated from the Dutch National Surgical Complication Registry (LHCR) with parallel assessment by the CDC definition through medical chart reviews. |

Continued

**Table 4** Continued

**WHO trial registration data set**

| Key secondary outcomes | ▶ SSI incidence evaluated at 30 days and 3 months follow-up by the CDC definition through medical chart review |
| | ▶ Readmissions rate within 30 days follow-up through LHCR registration and medical chart review |
| | ▶ All-cause mortality within 1 and 5 years after surgery |
| | ▶ WHO disability assessment schedule 2.0 by self-administration through online/paper-form questionnaires at postoperative day 30, 60 and 90 |
| | ▶ (In)direct medical and non-medical costs, quality-adjusted life years |
| | ▶ Incidence of anastomotic leakage at 30 days follow-up through LHCR registration and medical chart review |
| | ▶ Incidence of incisional hernia by medical chart review (diagnosed by either physical examination and/or ultrasonography or CT) 1 year after surgery |

CDC, Centers for Disease Control and Prevention; FiO$_2$, fractional inspired oxygen; SSI, surgical site infections.

open surgery as well as laparoscopic surgery with incisions larger than 5 cm, that is, for the excision specimen).

### Exclusion criteria

Subject meeting any of the following criteria will be excluded from participation in this study:
- ▶ The need for emergency surgery.
- ▶ Scheduled operation concerning a reoperation for complications from recent surgery (within 3 months after the initial procedure).
- ▶ Scheduled operation as part of a two stage surgical procedure and with second procedure possibly scheduled before follow-up of primary endpoint.
- ▶ The inability to read or understand informed consent material or study questionnaires.
- ▶ Participation in another study with interference of study interventions.
- ▶ Pregnancy.

### Interventions

The EPOCH intervention bundle and typical standard care are listed in table 1. Beyond this, standard SSI prevention measures typically include preoperative antibiotic prophylaxis when indicated, sterile technique, surgical site preparation and adherence to current national four-interventions SSI prevention (POWI) bundle consisting of hygiene discipline (measured by operating theatre door movements), timing of antibiotic prophylaxis (between 15 and 60 min prior to incision), normothermia (aimed at a core temperature of 36.5°C postoperatively) and the avoidance of preoperative hair removal. To avoid withholding good care, surgeons and anaesthesiologists can provide what they consider standard care at their discretion. However, it will be agreed that, whatever standard care is at the study initiation, this will not be structurally expanded to EPOCH interventions during the study period. We will aim to recruit study sites for collaboration that have not included the EPOCH interventions in their standard of care. In 2015, at the time of the study design

and initiation, none of the interventions that comprise EPOCH were regularly applied in Dutch hospitals.

Given that this is a pragmatic trial, patient care and comfort are prioritised at all times. Patient requests to cease any of the interventions are immediately honoured. Similarly, when a patient in the control group requests measures that would not have been withheld from them outside of the trial (ie, warming in case of discomfort from cold) these are also honoured. In case of iodine allergy, only chlorhexidine gluconate will be used for all aspects of surgical site handling. In case of chlorhexidine gluconate allergy, povidone iodine solution will be used. In the absence of allergy for either compound, chlorhexidine gluconate in alcohol is preferred for skin preparation.[32] For wound irrigation, the choice of agent is left to the discretion of the surgeon. In case of active atrial fibrillation, goal-directed therapy based on dynamic preload parameters is impossible. Therefore, hypotensive periods after induction are treated with a vasopressive agent (norepinephrine) until a maximum of 0.08 gamma and thereafter with fluids. Beyond this, there are no criteria for discontinuation or modification of the allocated intervention. The interventions included in EPOCH are individually recommended by the WHO and the CDC and are considered safe.[27–29] At all times, the care team is permitted to amend any of the interventions in both groups to guarantee safe anaesthesia and surgery.

Participation in trials that intervene with any of the interventions is not permitted. No other restrictions or recommendations are called for due to the pragmatic nature of the trial. Given the size of the study, any co-intervention that may affect the outcome is expected to randomise evenly to both treatment arms.

### Randomisation and treatment allocation

Patients are randomised per day, stratified by centre with a 1:1 ratio using variable block sizes to either intervention or control group with an internet based automated assignment system (Castor EDC).[33] The trial management team, or local study staff, will perform randomisation for

**Table 5** Roles and responsibilities in the EPOCH trial

**Roles and responsibilities in the EPOCH trial**

| Role | Details and responsibilities |
|------|------------------------------|
| Trial sponsor | Trial sponsor: Amsterdam University Medical Centre, Location AMC, Department of Surgery and Department of Anesthesiology<br>Principle Investigators: Professor dr. M.A. Boermeester and professor dr. dr. M.W. Hollmann<br>Address: Meibergdreef 9, 1105 AZ Amsterdam, The Netherlands<br>Email address: m.a.boermeester@amsterdamumc.nl, m.w.hollmann@amsterdamumc.nl<br>The sponsor conceived of the study and chairs the trial steering committee. The sponsor is not responsible for data collection, but is responsible for trial management, analysis, interpretations, writing of the report and decision to submit the report for publication. The sponsor does not provide any funding. |
| Principal investigator | The principal investigators (M.A. Boermeester and M.W. Hollmann) are responsible for the overall conduct of the trial. |
| Local investigators | The local investigator team consists of an anaesthesiologist and a surgeon at each trial site. The local investigators are responsible for local trial execution. |
| Trial Steering Committee (TSC) | The TSC consists of the principal investigators (M.A. Boermeester, Professor of Surgery and Clinical Epidemiologist; M.W. Hollmann, Professor of Anaesthesiology; M.G. Dijkgraaf, Professor of Health Technology Assessment) and a research physician (S.W. de Jonge, Clinical Epidemiologist). The TSC is responsible for design and overall conduct of the trial. This includes overseeing execution of the protocol, preparation of the protocol and potential amendments to the protocol, the statistical analysis plan, yearly progress reports to Amsterdam UMC medical ethics committee, safety reports to the Data and Safety Monitoring Board (DSMB) and the final report. |
| Study coordinator | The study coordinator (Q.J.J. Bolding and N. Wolfhagen) will coordinate the overall execution of the trial. This includes communication with local investigators, trial sites, if necessary included patients, and physicians executing the interventions. The study coordinator also oversees follow-up, data collection, data management and recruitment and start-up of new trials sites. |
| Trial management team | The trial management team (Q.J.J. Boldingh, N. Wolfhagen, S.W. de Jonge, W.J. Bom, L.M. Posthuma and J.C.G. Scheijmans) assists the study coordinator in the overall execution and if necessary assists local investigators with execution. The trial management team also performs screening of potential participants, recruitment of participants, randomisation and allocation of the intervention. |
| Data and Safety Monitoring Board | The DSMB safeguards the interests of trial participants and assesses the safety of the interventions during the trial. In addition, the DSMB performs the recalculation of the sample size. The DSMB consists of two clinicians with relevant subject matter knowledge (J.T.M. van der Meer, infectious disease specialist and M. Klimek, anaesthesiologist) and a statistician (P.M. Bossuyt). J.T.M. van der Meer chairs the DSMB. The DSMB reports to the trial sponsor. |
| Independent physician | The independent physician (T. Schepers, trauma surgeon) represents interests of trial participants and mediates conflicts between trial participants and investigators if necessary. |

each day and trial site after all potential participants for that day have decided on trial participation. This ensures allocation concealment.

Participation to the trial and treatment allocation is registered in a specific research section of the electronic medical record. Patients are randomised per day to avoid contamination between groups. Every day, an email will be sent to inform everyone involved in execution of the interventions, including anaesthesiologists, surgeons, operation assistants, recovery nurses and anaesthesia and ward nurses, of the treatment allocation for the day.

## Outcomes

The primary outcome measure is the difference in the proportion of patients developing a SSI between the two treatment arms as defined by the CDC definition within follow-up of 30 days, which will be evaluated from the Dutch National Surgical Complication Registry (LHCR) with parallel assessment through medical chart review by the trial management team to account for possible registration effect.[31] All study outcomes, their definitions and data collection methods are listed in table 2. Additionally, in a substudy, perioperative subcutaneous tissue oxygen pressure (PtO2) (mm Hg), mitochondrial PtO2

(mm Hg) and perioperative immune response will also be studied in a smaller population of 48 patients. Further details regarding these measurements and respective analysis is described in online supplementary appendix 1.

## Participant timeline

A schematic diagram of the participant timeline is depicted in figure 1.

## Sample size

Local SSI surveillance data (Unpublished: Amsterdam UMC, Location AMC; 2008 to 2012) show a 9.1% SSI risk with standard care. A maximum proportion of approximately 55% of SSI is deemed preventable by application of evidence-based strategies.[5] Accounting for a limited degree of contamination of intervention between groups and other factors contributing to SSI, not affected by the bundle, we deem a clinically relevant relative risk reduction of 30% reasonable. This 30% relative reduction represents a 9.1% SSI risk with standard care and a 6.37% SSI risk in the intervention group. A sample size of 3000 evaluable patients, 1500 in each arm, is sufficient to detect this 30% difference in proportion of SSI between the two groups using a two-tailed Z-test with 80% power and a 5% level of significance.

A report by the European Centre of Disease Prevention and Control indicates SSI incidence in the Netherlands to be considerably higher than 9.1% we used for our sample size estimation. We chose to use the more conservative value to prevent underestimation of the sample size. In addition, the randomisation per day per centre enables within-centre comparison, which is not yet accounted for in the current sample size estimation. Accounting for this may increase power or lower the number of patients required to attain the same power. Therefore, the independent statistician of the Data and Safety Monitoring Board (DSMB) will perform a sample size recalculation after inclusion of the first 1500 patients. This recalculation will account for within-centre effects and includes evaluation of the pre-study sample size assumption of the SSI incidence in the control group. As the primary outcome falls within regular follow-up visits, no missing data is expected. Therefore, the sample size is not corrected for missing data.

## Recruitment

At the study initiation, 11 hospitals committed to trial participation with a combined expected enrolment of 5419 participants over 33 months representing nearly 50% oversampling. Inclusion criteria are broad and straightforward and, given the low risk and potential benefit of the interventions, willingness to participate is expected to be high. No financial incentives are provided to investigators or participants.

## Blinding

Due to the nature of the interventions, trial participants and providers will not be blinded. Outcome assessors, ward doctors responsible for LHCR registration, are not involved in execution of the interventions and will be kept unaware of the treatment allocation.

## Data collection methods

All data will be collected and stored in a web-based electronic data capture system (Castor EDC).[33] Clinical data with regard to safety and efficacy outcomes will be collected manually by the trial management team during routine trial site visits using a standardised electronic case record form (eCRF). Baseline and clinical data will be collected through a detailed export directly from the electronic medical record to avoid human error. After data validation, data will be imported into the eCRF. Any modification to entered data is password secured and can only be processed with a corresponding explanation for the audit trail.

Survey data (from surgeon, anaesthesiologist and patients) will be collected via an eCRF on the EDC (electronic data capture). When email is not available, printed CRFs via surface mail will be used. A trial nurse will process surface mail responses in the eCRF. The trial management team will track survey responses, send reminders and conduct follow-up phone calls when necessary.

To monitor intervention adherence and potential contamination between groups, a postoperative survey is sent to surgeons and anaesthesiologists on execution of the EPOCH interventions and other interventions that may affect the outcome. In addition, relevant process measures (eg, administered $FiO_2$, measured temperature perioperative) are extracted from clinical data. For goal-directed fluid therapy and surgical site preparation we rely fully on the self-assessment checklist.

For the primary outcome, SSI is prospectively evaluated according to the CDC criteria.[31] SSI are registered by treating physicians as part of routine postoperative care and outpatient follow-up in accordance with the standardised Dutch National Surgical Complication Registration (LHCR) up to 30 days after surgery. Physicians involved are expected to be familiar with the CDC criteria. Physicians at the surgical ward are typically responsible for SSI registration but are not involved in the operative procedure. We do not expect operating surgeons to actively manipulate this registration and no additional measures are taken to formally exclude them from outcome assessment. Outside routine postoperative care, no additional follow-up visits are planned specifically for the trial. Timing of these follow-up visits vary and occur within 30 days after surgery. Patients are expected to return to the hospital whenever complications occur. If they do not, we will assume they did not develop a SSI. The research physicians from the trial management team will review all medical charts to account for a potential registration effect. Data collection on secondary outcomes is included in table 2.

## Retention and withdrawal

The burden of participation in the study will be minimised. With the exception of two glucose measurements

postoperatively and active preoperative and postoperative warming, the interventions take place while the patient is under anaesthesia. No active adherence from the patient is required and non-adherence from patients is therefore not expected. No additional visits to the hospital are required. Respondents are asked to complete four questionnaires over a period of 90 days. The trial management team will provide support and actively remind respondents via email, surface mail and phone calls. No (financial) incentives will be provided. Whenever a subject does decide to withdraw, all data collected up until withdrawal will be used.

The investigator can decide to withdraw a subject from the study for urgent medical reasons or when intraoperative findings indicate that inclusion criteria are not met anymore (eg, diagnostic laparoscopy, change in operative plan resulting in trans-anal approach), rendering the patient ineligible for the study. Patients withdrawn from the study are replaced.

### Data management
Subjects are coded by a numeric randomisation code. Local investigators are responsible for the subject identification log. Data are stored digitally and will be kept by the project leader for 15 years after the inclusion of the last patient.

### Statistical methods
All outcomes will be analysed according to the intention-to-treat principle. Binary outcomes will be analysed using log binomial regression to estimate relative risks. If this regression fails to converge, logistic regression will be performed using OR and corresponding CI for statistical interpretation and recalculated relative risk for point estimate interpretation. Continuous outcome data will be analysed using linear regression. Quality of life data will be analysed as repeated measurement using a linear mixed model. In all analyses, statistical uncertainties are expressed in 95% two-sided CIs. P values of <0.05 will indicate statistical significance.

Randomisation daily per trial site enables within-centre comparison of treatments and increases power. A sensitivity analysis of the primary outcome will be conducted that accounts for within-centre comparison of treatments. When the hypothesis test of this analysis diverges from the primary analysis, all further analysis will also be conducted using this effect estimate. All outcomes will also be analysed according to the per protocol population after adjustment for confounding due to incomplete adherence to the assigned treatments or use of off-protocol concomitant therapies. Variables will be considered for adjustment based on Tyler VanderWeele's principles for confounder selection and include preoperative body mass index, (insulin dependent) diabetes mellitus, cardiovascular heart diseases other than hypertension, chronic obstructive pulmonary disease and any other variables that meet these criteria and pass statistical variable selection.[34] Procedure duration is considered an important proxy for complexity of procedure and independent of the interventions comprising EPOCH. Therefore, it will also be considered for adjustment despite being measured during the exposure. In an additional analysis, this method will also be applied to each individual intervention of the EPOCH bundle, the POWI bundle and other promising interventions outside the bundle under investigation to assess the attributive effect of each intervention separately.

The intention-to-treat population will comprise of all evaluable randomised patients regardless of adherence of treatment allocation. The per protocol population will comprise of all patients who were treated per protocol according to the postoperative adherence surveys and process measures, and all patients in the control group who were not treated with these interventions according to the postoperative adherence surveys and process measures. The safety analysis will be conducted on all patients exposed to the interventions, regardless of evaluable primary outcome. Missing data will be handled by multiple imputation by chained equation when the required assumptions apply. A more detailed description of statistical methods and analyses is described in the statistical analysis plan (SAP). The EPOCH SAP will be published separate from the EPOCH Protocol timely before analysis of the data.

### Monitoring
An independent DSMB safeguards the interests of trial participants and assesses the safety of the interventions during the trial. The DSMB consists of a statistician, an anaesthesiologist and an infectious disease specialist. The DSMB will regularly monitor safety data, recruitment and loss to follow-up and make recommendations about continuation of the trial. After every 50 new serious adverse events (any type) or 10 new deaths, the DSMB will re-evaluate. If required, monitoring can also be on ad hoc basis. There will be no interim analysis of the treatment effect. At its own discretion, the DSMB may advice to stop the study prematurely based on its findings. This advice is not bound to specific guidelines other than that it should serve the interest of the trial participants. If the sponsor decides not to (fully) implement the advice, the sponsor will inform the MedicalEthics Committee (MEC).

### Harms
All serious adverse events (SAE) that occur within 30 days after surgery are recorded and reported to the DSMB irrespective of perceived relation to the intervention. SAE are defined as any event within 30 days of the intervention that is fatal, threatens the life of the subject, makes hospital admission or anextension of the admission necessary, causes persistent or significantinvalidity or work disability, manifests itself in a congenital abnormality ormalformation and/or could, according to the trial management team, havedeveloped to a serious undesired medical event, but was, however, prevented

dueto premature interference[35] . SAE will be followed up until they have abated, or until a stable situation has been reached.

All physicians involved in the EPOCH study are instructed to report serious adverse events to the study coordinator. Mortality has to be reported within 48 hours after the occurrence. In addition to spontaneous reporting, the trial management team will review medical charts within 30-day intervals. The MEC will receive a biannual line listing of all reported SAE.

In accordance with the Dutch human research law (WMO), the sponsor will suspend the study if there is sufficient ground that continuation of the study will jeopardise health or safety of study participants. The sponsor will notify the MEC without undue delay of a temporary halt including the reason for such an action. The study will be suspended pending a further decision by the MEC. When appropriate, the investigator will ensure that all subjects will be informed.

### Auditing
The Amsterdam UMC clinical research unit may decide to conduct audits and monitoring visits at random, independently of trial coordinator and sponsor. Due to minimal risks of the interventions there are no scheduled monitoring visits.

### Patient and public involvement
The Dutch patient and healthcare consumer federation (NPCF) endorsed the study and provided input on the study proposal.

### Rationale for the study design
An alternative to the present study design of 1:1 parallel group randomisation per day per trial site is cluster randomisation. Cluster randomisation can overcome the risk of contamination by randomising study participants by trial site. However, disadvantages of cluster randomisation include the risk of residual confounding due to imbalanced distribution of trial site characteristics (like baseline level of SSI prevention), and the need for a larger sample size to ensure equivalent statistical power.[36–38] This may occur despite large numbers of participants and clusters.[39] Substantial contamination can be tolerated within an individual randomised trial before a cluster design is better in terms of total sample size. By randomising per trial site per day, every patient treated at that particular site by that particular team will receive the same care during the entire day. At the individual patient level, the allocation remains completely random. This design foregoes the weaknesses of full cluster randomisation, while minimising the risk of contamination of usual care by the EPOCH enhanced care. Furthermore, contamination is taken into account in the sample size calculation.

## ETHICS AND DISSEMINATION
### Ethics
This study will be conducted in accordance with the Declaration of Helsinki (64th version, October 2013) and the laws governing human research in the Netherlands (Wet Medisch-wetenschappelijk Onderzoek met mensen – WMO and Best Clinical Practice (BCP)) and the guidelines of the Central Committee for Research involving Human Subjects (Centrale Commissie Mensgebonden Onderzoek - CCMO). The Amsterdam UMC MEC has approved this trial (reference 2015_121).

All intended protocol modifications with implications beyond administrative changes will be submitted to the MEC and only effectuated after formal approval for the change. Strictly administrative amendments will be notified to the MEC. Local principle investigators will be notified of all amendments. Participants will be notified if necessary. Current version of the protocol is V.10, 17 December 2019. All amendments are described in table 3.

### Confidentiality
Informed consent forms will be securely stored at local study sites in locked cabinets with limited access. All data will be linked to an anonymised study number. The local investigators are responsible for the subject identification log. All local databases will be secured by password. All other information regarding the trial will be stored securely in a local trial master file. Data collected during this trial will only be used for the objective of this study described in the informed consent unless additional consent of the participant is retrieved. The informed consent form is enclosed in online supplementary appendix 2. The trial steering committee and study coordinator will control access of the final data set.

### Ancillary and post-trial care
No provisions for post-trial care are made. MEC has waived the obligation to take out a special insurance for participants because there are no risks associated with participation to this study.

### Dissemination
No arrangements have been made concerning public disclosure. The study is registered in the Dutch trial register as Trial NL5572 (https://www.trialregister.nl/trial/5572). An overview of the WHO trial registration data set is described in table 4. The trial results will be submitted to a peer-reviewed journal regardless of the outcome and made open access available in accordance with the Netherlands Organisation for Health Research and Innovation (ZonMW) policy. Co-authorship will be based on the international committee of medical journal editors' guidelines. Contributors that do not fulfil these criteria will be listed as collaborator. The order of authors will be based on scientific input.

The study protocol and statistical analysis plan will be published after half of the planned sample size, 1500 evaluable participants, have been enrolled and before any of

the outcome data will be available. Participant level data and statistical code will be made available on request after all trials results have been published and a data sharing agreement has been establishment.

## Roles and responsibilities
Roles and responsibilities are described in table 5.

**Author affiliations**
[1]Department of Surgery, Amsterdam UMC - Locatie AMC, Amsterdam, Noord-Holland, Netherlands
[2]Department of Anesthesiology, Amsterdam UMC - Locatie AMC, Amsterdam, Noord-Holland, Netherlands
[3]Department of Anesthesiology, Albert Schweitzer Hospital, Dordrecht, Noord-Holland, Netherlands
[4]Department of Surgery, Albert Schweitzer Hospital, Dordrecht, Zuid-Holland, Netherlands
[5]Anesthesiology, Dijklander Ziekenhuis, Hoorn, Noord-Holland, Netherlands
[6]Department of Surgery, Dijklander Ziekenhuis, Hoorn, Noord-Holland, Netherlands
[7]Department of Anesthesiology, Tergooiziekenhuizen, Hilversum, Noord-Holland, Netherlands
[8]Department of Surgery, Tergooiziekenhuizen, Hilversum, Noord-Holland, Netherlands
[9]Department of Anesthesiology, Rode Kruis Ziekenhuis, Beverwijk, Noord-Holland, Netherlands
[10]Department of Surgery, Rode Kruis Ziekenhuis, Beverwijk, Noord-Holland, Netherlands
[11]Department of Gynaecologic Oncology, Amsterdam UMC - Locatie AMC, Amsterdam, Noord-Holland, Netherlands
[12]Department of Anesthesiology, Jeroen Bosch Hospital, 's-Hertogenbosch, Noord-Brabant, Netherlands
[13]Department of Surgery, Jeroen Bosch Ziekenhuis, 's-Hertogenbosch, Noord-Brabant, Netherlands
[14]Department of Anesthesiology, Maasstad Ziekenhuis, Rotterdam, Zuid-Holland, Netherlands
[15]Department of Surgery, Maasstad Ziekenhuis, Rotterdam, Zuid-Holland, Netherlands
[16]Department of Anesthesiology, Gelre Ziekenhuizen, Apeldoorn, Gelderland, Netherlands
[17]Department of Surgery, Gelre Ziekenhuizen, Apeldoorn, Gelderland, Netherlands
[18]Clinical Epidemiology, Biostatistics and Bioinformatics, Amsterdam UMC - Locatie AMC, Amsterdam, Noord-Holland, Netherlands

**Contributors** MB and MWH conceived of the study. MB, MWH, MD and SJ designed the study. MD provided statistical and methodological expertise. MB provided expertise from the surgical perspective and MWH provided expertise from the anaesthesiological perspective. MB, MWH, MD, SJ, QB, LP, WB, NW and JS contributed to the study protocol. All local investigators (BL, JH, JPH, DS, OG, ER, EK, AD, LL, TS, WD, JK, CR, LO, PD) provided essential critical input on the study protocol. MB guided and supervised SJ with the grant application, obtaining medical ethics approval and writing of the study protocol. SJ, QB, LP, WB, NW and JS will help with implementation and execution of the trial. SJ and NW will conduct the statistical analysis supervised by MD. SJ, NW and QB will draft the final manuscript under supervision of MB, MD and MWH.

**Funding** The EPOCH trial is funded by the Netherlands Organization for Health Research and Innovation (ZonMw; project 843002606) with 10% compulsory co-financing provided by the Innovation Fund of Dutch healthcare insurers and by Ethicon (Johnson & Johnson).

**Competing interests** None declared.

**Patient and public involvement** Patients and/or the public were involved in the design, or conduct, or reporting or dissemination plans of this research. Refer to the Methods section for further details.

**Patient consent for publication** Not required.

**Provenance and peer review** Not commissioned; externally peer reviewed.

**Data availability statement** Data sharing not applicable as no data sets generated and/or analysed for this protocol.

**ORCID iD**
Niels Wolfhagen http://orcid.org/0000-0003-0573-5703

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
