## [Reviewer comments · BMJ Open]

ARTICLE DETAILS

TITLE (PROVISIONAL)	Enhanced PeriOperative Care and Health protection program for the prevention of surgical site infections after elective abdominal surgery (EPOCH); study protocol of a randomised controlled multicentre superiority trial
AUTHORS	de Jonge, Stijn; Wolfhagen, Niels; Boldingh, Quirine; Bom, Wouter; Posthuma, Linda; Scheijmans, Jochem; van der Leeuw, Bart; van der Hoeven, Joost; Hering, Jens Peter; Sonneveld, Dirk; van Geffen, Otto; Hendriks, Eduard; Kluyver, Ewoud; Demirkiran, Ahmet; van Lonkhuijzen, Luc; Slotema, Thomas; Draaisma, Werner; Koopman, Seppe; van Rossem, Charles; Over, Linda; van Duijvendijk, Peter; Dijkgraaf, Marcel; Hollmann, Markus; Boermeester, Marja

VERSION 1 – REVIEW

REVIEWER	Colin Peirce University Hospital Limerick, Limerick, Ireland
REVIEW RETURNED	24-Mar-2020

GENERAL COMMENTS	Thank you - this will be a very valuable study in due course. I have only a few comments: 1. In the Interventions section, the authors say 'in case of iodine allergy, chlorhexidine will be used etc.'. This sentence is a little confusing when compared to surgical site handling section of Table 1. Might be omitted? Or else clarify in text that chlorhexidine is the preparation of choice for skin and iodine for wound irrigation (in Interventions section).2. The patients in EPOCH arm it would seem are far more likely to have an arterial catheter placed for monitoring - this has its own potential morbidity, will this be followed?3. All cases, benign and malignant to be included. I note in the Appendix for tissue oxygen measurements, the 48 patients included would be excluded if on steroids, immunosuppressants, chemotherapy etc. What is the rationale for not excluding or indeed including these in the whole study cohort as we know that these factors contribute to increased likelihood of developing SSI?4. I do not work in the Dutch healthcare system and don't know what the LHCR entails. Can it be explained exactly when, where and by whom the patient will be assessed postoperatively for SSI? Are the operating surgeons involved in this process?5. To save me from retrieving references 35 and 36, are incisional hernias a clinical diagnosis? Or radiological? Or both? If radiology utilised, what method and this will be performed 12 months postoperatively for all patients?6. The calculations are based on SSI incidence of 9.1% over period
---

	2008-2012 i.e. 8 to 12 years pre-trial commencement. Is there a more recent figure for SSI incidence available? 7. In the submitted chronology of the trial, I think point 3 has the incorrect date (2019 listed). Many thanks again and I look forward to reading the results in due course.
--	---

REVIEWER	markus diener Department of General, visceral and Transplantation Surgery, Heidelberg, Germany
REVIEW RETURNED	03-Apr-2020

GENERAL COMMENTS	Dear authors, dear Editors, this study protocol is outstanding and after multiple attempts to reduce SSI by single measures it is the Right way to go with bundle, Composite interventions. Please allow me just some minor comments: Introduction: with 1200 randomised patients, the PROUD Trial represents a strong piece of evidence for SSI; as the applied bundle uses evidence based Intervention non-effectiveness of Triclosan coated sutures should be mentioned and referenced in the introduction, since this was and is a big matter of scientific (and industrial) debate. -Methods/sample size: without going into Details of underlying evidence the Risk rates seem to be optimistic, from my Point of view current SSI rates (assessed in proper Trials) lie beyond 10% margin, will potentially Impact the significance if a priori guess is wrong -please describe the Methods of Primary endpoint assessment in Detail: telephone interview, necessity of photo documentation by Patient of the wound? will the Patient be seen in Hospital in case of uncertainty etc -important study! good luck
--

VERSION 1 – AUTHOR RESPONSE

Reviewer 1

General We thank the reviewer for the careful review of our manuscript and for these thoughtful response suggestions

Comment #1 In the Interventions section, the authors say 'in case of iodine allergy, chlorhexidine will be used etc.'. This sentence is a little confusing when compared to surgical site handling section of Table 1. Might be omitted? Or else clarify in text that chlorhexidine is the preparation of choice for skin and iodine for wound irrigation (in Interventions section).

Response We provided clarification of this statement, line 174 - 178:

Main Document – marked copy, Methods and analysis, Page 8, Line 174 - 178:
"In case of iodine allergy, only chlorhexidine gluconate will be used for all aspects of surgical site handling. In case of chlorhexidine gluconate allergy, povidone iodine solution will be used. In the absence of allergy for either compound, chlorhexidine gluconate in alcohol is preferred for skin preparation. 32 For wound irrigation, the choice of agent is left to the discretion of the surgeon."

32WHO. WHO Surgical Site Infection Prevention Guidelines; Webappendix 8; Summary of a systematic literature review on surgical site preparation [Available from: <https://www.who.int/gpsc/appendix8.pdf> accessed 14-04-2020.

Comment #2 The patients in EPOCH arm it would seem are far more likely to have an arterial catheter placed for monitoring - this has its own potential morbidity, will this be followed

Response Clinically relevant complications of arterial catheter placement fall within the study definition of serious adverse events* and will be followed up as such. However, we do not anticipate differences in the frequency of arterial catheter use between the two groups as the study protocol does not call for arterial catheter placement. Arterial catheters will be used for goal directed fluid therapy only when clinically indicated.

Arterial catheters will not be placed solely for the trial. Alternative approaches for goal directed fluid therapy without arterial catheter are available for patients for whom arterial catheters are not clinically indicated. To clarify an additional statement is added in table 1.

Main Document – marked copy, Methods and analysis, Page 8, Table 1:

“Goal directed fluid therapy: Patients will receive hemodynamic therapy based on a goal directed approach. Optimisation is preferably based on dynamic pre-load parameters (pulse pressure variation or systolic pressure variation) derived from arterial catheter measurements, when such catheter is used based on clinical indication. Arterial catheters will not be placed solely for the trial. When no arterial catheter is indicated, a goal directed approach using a non-invasive hemodynamic monitoring system may be used. When both an arterial catheter and a non-invasive hemodynamic monitoring system are not available, hypotensive periods following induction should be compensated for by employing a vasopressor agent (norepinephrine) until a maximum of 0.08 gamma before fluids are administered.”

*definition of serious adverse events: any event that results in death, is life threatening (at the time of the event), requires prolonged hospitalisation or re-hospitalisation, results in persistent or significant disability or incapacity, is a congenital anomaly or birth defect, or any other event that may jeopardise the subject or require an intervention to prevent one of the outcomes listed before.

Comment #3 All cases, benign and malignant to be included. I note in the Appendix for tissue oxygen measurements, the 24 patients included would be excluded if on steroids, immunosuppressants, chemotherapy etc. What is the rationale for not excluding or indeed including these in the whole study cohort as we know that these factors contribute to increased likelihood of developing SSI?

Response The sub-study, aimed to explore the underlying mechanism of a potential effect, is limited in size due to the considerable costs involved with the additional measurements. Forty-eight patients will undergo tissue oxygen tension measurement, but also other

measurements including a quantification of the perioperative immune response by determination of circulating cytokines. Immunosuppression blunts cytokine production. Potential differences, if any, would not be detectable in patients using these agents. To ensure that the two relatively small study arms are comparable, these patients are excluded from the sub-study.

In the larger study population, we expect that any of these patient characteristics would randomise equally across the two study arms and therefore not affect the overall effect estimate of the intervention. For clarification an additional line is added in Appendix 1:

Appendix 1, Page 1:

“As immunosuppressants blunt immunologic effects, potential differences, if any, will not be detectable in these patients. Therefore, patients using immunosuppressants are excluded for these additional measurements.”

The sub- study, after protocol V9, concerns 48 patients. Therefore another adjustment was made in line 211:

Main Document – marked copy, Methods and analysis, Page 10, Line 209-211: “Additionally, in a sub-study, perioperative subcutaneous PtO₂ (mmHg), mitochondrial PtO₂ (mmHg) and perioperative immune response will also be studied in a smaller population of 48 patients.”

Comments #4 I do not work in the Dutch healthcare system and don't know what the LHCR entails. Can it be explained exactly when, where and by whom the patient will be assessed postoperatively for SSI? Are the operating surgeons involved in this process?

Response We understand readers may be unfamiliar with the LHCR and added clarification on how, when and by whom follow-up of SSI is planned.

Main Document – marked copy, Methods and analysis, Page 13, Line 276-286: "SSI are registered by treating physicians as part of routine postoperative care and outpatient follow-up in accordance with the standardised Dutch National Surgical Complication Registration (LHCR) up to 30 days after surgery. Physicians involved are expected to be familiar with the CDC criteria. Physicians at the surgical ward are typically responsible for SSI registration but are not involved in the operative procedure. We do not expect operating surgeons to actively manipulate this registration and no additional measures are taken to formally exclude them from outcome assessment. Outside routine postoperative care, no additional follow-up visits are planned specifically for the trial. Timing of these follow-up visits vary and occur within 30 days after surgery. Patients are expected to return to the hospital whenever complications occur. If they do not, we will assume they did not develop a SSI. The research physicians from the trial management team will review all medical charts to account for a potential registration effect."

Comment #5 To save me from retrieving references 35 and 36, are incisional hernias a clinical diagnosis? Or radiological? Or both? If radiology utilised, what method and this will be performed 12 months postoperatively for all patients?

Response Medical records will be checked to see if patients have had a (clinical) diagnosis of incisional hernia and required new surgery. Patients will not receive additional diagnostics as follow-up. We added wording to clarify this to the definition in table 3:

Main Document – marked copy, Methods and analysis, Page 10, Table 2: "Incisional hernia defined by the European Hernia Society, as any abdominal wall gap with or without bulge in the area of a postoperative scar perceptible or palpable by clinical examination or imaging"

Comment #6 The calculations are based on SSI incidence of 9.1% over period 2008-2012 i.e. 8 to 12 years pre-trial commencement. Is there a more recent figure for SSI incidence available?

Response We appreciate the reviewer's concern regarding the SSI incidence. Preparations for the trial, including sample size estimation, and search for funding started in 2013. At the time, the figure was timely. To avoid further delay but accommodate adaptation of the sample size to the most recent and reliable incidence, we incorporated a sample size recalculation after inclusion of 1500 patients. The incidence of SSI in the control group of these 1500 patients will be used to recalculate the sample size. The independent statistician of the Data Safety Monitoring Board will perform this recalculation. We added wording to the manuscript to clarify this:

Main Document – marked copy, Methods and analysis, Page 12, Line 236-239:

"Therefore, the independent statistician of the Data Safety Monitoring Board (DSMB) will perform a sample size recalculation after inclusion of the first 1500 patients. This recalculation will account for within-centre effects and includes evaluation of the pre-study sample size assumption of the SSI incidence in the control group."

Comment #7 In the submitted chronology of the trial, I think point 3 has the incorrect date (2019 listed).

Response The date is revised into 2015

Reviewer 2

General We thank the reviewer for the careful review of our manuscript, kind words and for these response thoughtful suggestions.

Comment #1 Introduction: with 1200 randomised patients, the PROUD Trial represents a strong piece of evidence for SSI; as the applied bundle uses evidence based Intervention non-effectiveness of Triclosan coated sutures should be mentioned and referenced in the introduction, since this was and is a big matter of scientific (and industrial) debate.

Response We have added a reference to the trial in the introduction where individual trials without clinically relevant SSI reduction are mentioned. We greatly value this work and acknowledge the importance to the field. At the time of the first draft of our introduction in the original study protocol, the PROUD trial was not yet published.

Main Document – marked copy, Background and rationale, Page 5, line 88-89:

"However, individual measures alone have not resulted in a clinically relevant SSI reduction. 6-9 "

Comment #2 Methods/sample size: without going into Details of underlying evidence the Risk rates seem to be optimistic, from my Point of view current SSI rates (assessed in proper Trials) lie beyond 10% margin, will potentially Impact the significance if a priori guess is wrong.

Response We recognise the importance of an accurate up-to-date estimate of current SSI incidence. When the study was designed, local SSI surveillance indicated a SSI risk of 9.1%, but indications that the real incidence may be higher did exist. We chose to use the more conservative approach (i.e. leading to the largest sample size) but planned a sample size recalculation by the independent statistician of the Data Safety Monitoring Board after 1500 inclusions to account for a potentially higher incidence in the control group. This is described in the Sample size section on page 12. We have added wording to clarify the process:

Main Document – marked copy, Methods and analysis, Page 12, Line 236-239:
 “Therefore, the independent statistician of the Data Safety Monitoring Board (DSMB) will perform a sample size recalculation after inclusion of the first 1500 patients. This recalculation will account for within-centre effects and includes evaluation of the pre-study sample size assumption of the SSI incidence in the control group.”

Comment #3

Please describe the Methods of Primary endpoint assessment in Detail: telephone interview, necessity of photo documentation by Patient of the wound? will the Patient be seen in Hospital in case of uncertainty etc

Response We understand the need for more detail regarding follow-up of the primary outcome. A more elaborate explanation is now given in the manuscript.

Main Document – marked copy, Methods and analysis, Page 13, Line 276-286:
 “SSI are registered by treating physicians as part of routine postoperative care and outpatient follow-up in accordance with the standardised Dutch National Surgical Complication Registration (LHCR) up to 30 days after surgery. Physicians involved are expected to be familiar with the CDC criteria. Physicians at the surgical ward are typically responsible for SSI registration but are not involved in the operative procedure. We do not expect operating surgeons to actively manipulate this registration and no additional measures are taken to formally exclude them from outcome assessment. Outside routine postoperative care, no additional follow-up visits are planned specifically for the trial. Timing of these follow-up visits vary and occur within 30 days after surgery. Patients are expected to return to the hospital whenever complications occur. If they do not, we will assume they did not develop a SSI. The research physicians from the trial management team will review all medical charts to account for a potential registration effect.”

VERSION 2 – REVIEW

REVIEWER	Colin Peirce University Hospital Limerick, Limerick, Ireland
REVIEW RETURNED	21-Apr-2020
GENERAL COMMENTS	Many thanks for your replies which are satisfactory and well explained. I look forward to hearing how this trial progresses and the results in due course.